# Comparative Analysis of Dasatinib Effect between 2D and 3D Tumor Cell Cultures

**DOI:** 10.3390/pharmaceutics15020372

**Published:** 2023-01-21

**Authors:** Samantha Sabetta, Davide Vecchiotti, Letizia Clementi, Mauro Di Vito Nolfi, Francesca Zazzeroni, Adriano Angelucci

**Affiliations:** Department of Biotechnological and Applied Clinical Science, University of L’Aquila, 67100 L’Aquila, Italy

**Keywords:** 3D tissue culture, 3D tumor model, tumor spheroids, drug resistance, drug screening, preclinical study

## Abstract

Three-dimensional cell culture methods are able to confer new predictive relevance to in vitro tumor models. In particular, the 3D multicellular tumor spheroids model is considered to better resemble tumor complexity associated with drug resistance compared to the 2D monolayer model. Recent advances in 3D printing techniques and suitable biomaterials have offered new promises in developing 3D tissue cultures at increased reproducibility and with high-throughput characteristics. In our study, we compared the sensitivity to dasatinib treatment in two different cancer cell lines, prostate cancer cells DU145 and glioblastoma cells U87, cultured in the 3D spheroids model and in the 3D bioprinting model. DU145 and U87 cells were able to proliferate in 3D alginate/gelatin bioprinted structures for two weeks, forming spheroid aggregates. The treatment with dasatinib demonstrated that bioprinted cells were considerably more resistant to drug toxicity than corresponding cells cultured in monolayer, in a way that was comparable to behavior observed in the 3D spheroids model. Recovery and analysis of cells from 3D bioprinted structures led us to hypothesize that dasatinib resistance was dependent on a scarce penetrance of the drug, a phenomenon commonly reported also in spheroids. In conclusion, the 3D bioprinted model utilizing alginate/gelatin hydrogel was demonstrated to be a suitable model in drug screening when spheroid growth is required, offering advantages in feasibility, reproducibility, and scalability compared to the classical 3D spheroids model.

## 1. Introduction

Inhibition of key molecules in signaling pathways that regulate cancer cell proliferation and dissemination is one of the most promising strategies in modern cancer therapy. Specifically, tyrosine kinases represent a particularly important target because they mediate several biological processes, including cell growth and differentiation, metabolism, and apoptosis. Constitutive activation gives these enzymes an oncogene status, which directs the cell toward neoplastic transformation [1].

Src, a 60-kDa protein, is the most widely studied member of the Src family tyrosine kinases (SFKs), being frequently overexpressed in many tumors. SFKs are a large class of non-receptor kinases that control multiple signaling pathways in animal cells, the activation of which is necessary for mitogenic signaling of many growth factors, but also for the acquisition of the migratory and invasive phenotype. Indeed, oncogenic activation of SFKs has been shown to play an important role in solid tumors, promoting tumor growth and distant metastasis formation, and that is why they represent the most representative class of targeted proteins in cancer therapy [2]. This family consists of 11 highly homologous members: c-Src, Blk, Fgr, Fyn, Frk, Hck, Lck, Lyn, Yes, Yrk, and Srms [3,4]. Of the 11 members of SFKs, Src, Fyn, and Yes are implicated in tumorigenesis and metastasis formation [5]. In fact, Src, Fyn, and Yes, but also Frk, are widely expressed in a variety of tissues, while for the other members, protein expression is more restricted in tissues, with a prevalence in cells of hematopoietic origin [6].

Src activity is regulated, at the cellular level, by the phosphorylation state of two key tyrosine residues and by interactions with ligands mediated by its regulatory domains, which stabilize the kinase in active or inactive configuration. In the inactive configuration, the SH2 domain binds phosphorylated Tyr527 in the C-terminal domain, while the SH3 domain interacts with the linker domain on the backside of the catalytic domain, thus promoting a closed conformation that prevents interaction with substrates. In the active configuration, Tyr527 is dephosphorylated, and the SH2 and SH3 domains are released by intramolecular interactions and thus available for binding of heterologous molecular partners; this open conformation also allows Tyr416 to be self-phosphorylated.

In cancer cells, Src transmits signals that promote cell survival and mitogenesis; in addition, Src exerts a profound effect on cytoskeleton reorganization and adhesion systems that underlie cell migration and invasion. Src, therefore, not only promotes cancer cell growth but is also involved in the control of adhesiveness and migration, functioning as a key molecule that regulates signal transduction pathways triggered by various surface molecules, such as growth factor receptors and integrins [2,7]. Indeed, most evidence suggests that Src has a predominant role in the maintenance of the neoplastic phenotype and tumor progression, rather than tumor initiation or growth [8]. During tumor progression, Src activity becomes abnormally elevated, and because mutation activation or amplification of Src is very rare in human tumors, altered extrinsic control of Src phosphorylation by kinase or phosphatase may represent an important mechanism of Src upregulation.

Dasatinib (BMS-354825, Sprycel) is an orally available small molecule that in 2006 received FDA approval for the treatment of chronic myelogenous leukemia and Philadelphia-positive acute lymphoblastic leukemia. Dasatinib has a low inhibition specificity and its potential targets comprise Src in addition to BCR-Abl, c-kit, PDGFR, and other SFKs, including Lck, Fyn, and Yes, with IC50 < 1.0 nmol/L [9]. Preclinical data suggest that dasatinib could be effective also in solid cancer, and numerous results from phase I/II trials conducted in different cancer types have been published since 2009 [10]. Despite the encouraging preclinical results, the clinical validation of these data has been largely unproductive. In fact, although in vitro studies have demonstrated several gene signatures predictive of dasatinib response, these signatures have not yet permitted the definition of cohorts clinically sensitive to dasatinib [11]. In addition, available preclinical models have been shown to be largely inadequate in their predictive capacity, and new experimental strategies implementing microenvironment condition are needed. In fact, SFK activity is tightly dependent on ECM composition, and a changeable microenvironment could heavily impact SFK functions and thus treatment efficacy.

The tumor microenvironment is known to be complex, both in its content and dynamic nature, which is difficult to study using two-dimensional (2D) cell culture models. In contrast, three-dimensional (3D) cell models, such as spheroids, show peculiar molecular features that differ from monolayer cultures but are closer to the structural architecture of the tumor in vivo. Such models, in fact, can more accurately mimic both the structure of the malignant tissue and its microenvironment (physiological responses, secretion of soluble mediators, gene expression, patterns and mechanisms of drug resistance). Tumor cells within the spheroid reproduce the same concentric arrangement as tumors; in fact, they have a proliferative outer layer, a quiescent middle layer, and a hypoxic and necrotic core and have a growth pattern like solid tumors in the early non-vascularized stage. The use of 3D cell assays adds value to research and screening studies to identify potential anticancer compounds, bridging existing limitations between 2D cell cultures and animal models.

The theoretic superiority of 3D vs. 2D cultures has prompted the development of several protocols with the purpose to overcome the main weaknesses of 3D tissue cultures: technical complexity in performing and analyzing, high costs, reproducibility, and scalability. In particular, different approaches have been described to realize 3D sphere culture. Hanging drop protocol is a suspension culture method and it is one of the most popular approaches used for culture of 3D spheres, because it does not need particular or expensive instruments [12]. There are other suspension culture approaches for large-scale formation of spheres, each with their distinct advantages and disadvantages [13]. They include device-assisted culture, in order to force cell aggregation by movement or magnetic levitation, and gel embedding culture, able to mimic ECM microenvironment. In particular, the use of gels or scaffolds may guarantee a higher predictivity of the tissue models because cells organize a complex dynamic 3D architecture making contact also with ECM [14].

Three-dimensional bioprinting is an emerging technology that, using dedicated software and hardware to design 3D patterns and structures, aims to precisely produce an engineered structure or tissue that can be used for biological and pharmacological studies or directly in humans for regenerative medicine. It represents a relatively new approach that provides high reproducibility and precise control over automated fabricated constructs, potentially enabling high-throughput production of the desired 3D model. Bioprinting modalities include extrusion-based bioprinting [15], inkjet-based bioprinting [16], and laser-based bioprinting. Extrusion-based bioprinting is the most common and reliable system for 3D printing of biological tissues and is characterized by robotic delivery of a continuous stream of bioink, under pneumatic or motorized forces.

During the bioprinting process, a solution of a biomaterial or a mixture of several biomaterials in the form of a hydrogel, which usually encapsulates the desired cell types, called a bioink, is used to create tissue scaffolds with the final shape resembling the designed construct. Bioinks can be made alone from natural or synthetic biomaterials or from a combination of the two as hybrid materials. Hydrogel is the most widely used bioink, as it has properties like those of extracellular matrices and also allows the encapsulation of cells in a 3D environment that is both highly hydrated and mechanically stable. Sodium alginate is frequently used as hydrogel in 3D tissue culture thanks to its favorable physio-chemical characteristics, including gelling, viscous, and stabilizing properties, and the ability to retain water. Although sodium alginate is biocompatible and biodegradable, it is frequently mixed with other polymers of animal origin in order to enhance its gelling properties and the viability of encapsulated cells. The combination of sodium alginate and gelatin provides an excellent hydrogel for use as a substrate in 3D printing technology due to its biological properties, such as its biocompatibility, biodegradability, and non-toxicity. The easily modified mechanical properties of these materials can be adapted to living tissue, making them ideal environments for cell culture development. However, the chemical and mechanical cues provided by the specific hydrogel formulation as well as protocol used when performing 3D culture can affect the behavior of spheres and their resistance to drugs, requesting more confirmatory data [17,18].

In our study, we aimed to investigate the preclinical potentiality of 3D bioprinting technology in the field of experimental oncology. In particular, we verified the effect of dasatinib in 3D bioprinting models based on alginate/gelatin matrix utilizing two different tumor cell models, prostate cancer and glioblastoma, in comparison with the corresponding 3D spheroids models and 2D monolayer cultures.

## 2. Material and Methods

### 2.1. Cell Lines and 2D Monolayer Culture

Experiments were performed using U87 MG cell line (U87) human glioblastoma astrocytoma derived from a malignant glioma from a female patient by the explant technique, able to produce a malignant tumor consistent with glioblastoma in nude mice, and DU145 cell line human prostate adenocarcinoma isolated from the brain of a 69-year-old male. The U87 cells were cultured in DMEM high-glucose growth medium and DU145 cells in RPMI 1640 medium. For all cell lines, the growth medium was supplemented with 10% fetal bovine serum, 2 mM glutamine 100 IU/mL penicillin, and 100 μg/mL streptomycin. Cell lines were supplied by ECACC and underwent regular testing for mycoplasma by Hoechst DNA staining and PCR. Authentication procedures included species verification by DNA barcoding and identity verification by DNA profiling. U87 and DU145 cells when passed were initially plated at a density of 10^4^ cells/cm^2^, and incubated in 5% CO_2_ at 37 °C. For protein extraction, we initially recovered the cells with the scraper and then centrifuged at 300× *g* for 10 min and then at 100× *g* for 5 min. Cells were processed for protein extraction with cell lysis buffer containing 0.1% Triton X-100, 10 mM Tris-HCl (pH 7.5), 150 mM NaCl, 5 mM EDTA, and supplemented with 1 mM Na_3_VO_4_ and 75 U of aprotinin (Sigma-Aldrich, St. Louis, MI, USA), and incubated for 20 min at 4 °C. Protein extracts were stored at −80 °C until use.

### 2.2. Preparation of Hydrogel and Bioprinting

The hydrogel used for bioprinting is composed of 2% alginate and 8% gelatin (bioink). The powders were initially exposed to UV light for 15 min and then dissolved in sterile DPBS on a magnetic stirrer at 50 °C under laminar flow. Once prepared, bioink was placed in sterile syringes and stored at 4 °C until use. Prior to the bioprinting procedure, bioink was equilibrated at 37 °C. Next, 20 × 10^6^ cells were suspended in 400 μL of culture medium and then mixed with 3.7 mL of bioink to reach a final concentration of 5 × 10^6^ cell/mL. Finally, a disposable cartridge was filled with cell-laden hydrogel and equilibrated at 29 °C for 30 min in a temperature-controlled printhead. Printing was carried out using BIO X^TM^ bioprinter (Cellink, Gothenburg, Sweden) in a 12-well plate with a dispensing pressure of 50 kPa at a speed of 5 mm/s and setting the printbed temperature at 18 °C. The 3D structures were finally cross-linked with CaCl_2_ for 10 min and for an additional 3 min with BaCl_2_ before adding culture medium. For protein extraction, the 3D structures were quickly dissolved in a solution of EDTA 250 mM in PBS and cells were collected by centrifugation at 400× *g* for 10 min. The pellet was resuspended in PBS and centrifugated at 400× *g* for 5 min. The resulting pellet was dissolved in RIPA buffer for protein extraction.

### 2.3. Cell Viability and IC50 Analysis

DU145 and U87 cell lines were seeded at a density of 5 × 10^4^ cells/mL in a 24 multiwell. After 24 h, the cells were treated with dasatinib by the serial dilution method for 72 h. At the end of 72 h, the growth medium was removed and substituted with fresh medium. Cell viability in both 2D and 3D cultures was assessed using PrestoBlue^®^ (Thermo Fisher, Waltham, MA, USA) colorimetric viability assay, based on a ready-to-use, non-toxic, cell-permeable resazurin-based solution. It functions as a cell viability indicator by using the reducing power of living cells and quantitatively measuring the proliferation of cells. Cells were incubated with reagent in a 1:10 ratio for 2 h, in the dark. Then, a spectrophotometer reading was taken at wavelengths of 570 and 600 nm. For IC50 analysis, data collected in triplicate were elaborated with DRFit software (http://www.structuralchemistry.org/pcsb/drfit.php, accessed on 2 November 2022).

### 2.4. Western Blot

Protein extracts were centrifuged for 10 min at 300× *g* to eliminate nuclei and large debris. After protein dosage by Bradford Dye Reagent assay (Bio-Rad, Hercules, CA, USA), the same protein quantity for each sample was subjected to 10% sodium-dodecyl sulphate polyacrilamide gel electrophoresis (SDS-PAGE). Prestained protein molecular markers sharpmass VII (Euroclone, Milan, Italy) were loaded on a separate well for each gel. Then, at the end of the run, proteins were electrophoretically transferred from gel onto nitrocellulose membranes Amersham protran 0.2 μm (Cytiva Europe, Freiburg, Germany) for 90 min at 350 mA. Membranes were incubated for 1 h at RT with 10% nonfat dry milk in Tris-buffered saline (Bio-Rad) at pH 7.4 containing 20 mM Tris, 500 mM NaCl, and supplemented with 0.05% Tween 20 (Bio-Rad), and then probed for 1 h at RT with primary antibodies according to dilution suggested by the manufacturer: anti-Src (36D10), anti-Phospho-Src (Thr416) (E6G4R), and anti-GAPDH (0411) (all from Cell Signaling Technology, Danvers, MA, USA). Protein bands were visualized after 1 h of incubation with horseradish peroxidase (HRP)-conjugated anti-rabbit IgG or anti-mouse IgG (Cell Signaling Technology) at RT, and then with chemiluminescence reagents (Amersham, Buckinghamshire, UK). Chemiluminescent signals were acquired by the Chemidoc XRS system and digitally analyzed for the determination of band molecular weight and density by Imagelab software (Bio-Rad).

### 2.5. 3D Multicellular Spheroid Model

For the formation of U87 and DU145 spheroids, we used the “hanging drops” method. The density chosen was 500,000 cells/mL, which were resuspended in 10 μL of growth medium with 25% FBS and 25 μL of collagen (10 μg/mL). This method involves the deposition of 1 μL of cells on the bottom of the lid of a Petri dish; inverting the dish forms a “hanging” drop, and the cells, by gravity, accumulate in the bottom of the drop, promoting cell aggregation into spheroids. The spheroids were kept in an incubator at 37 °C and 5% CO_2_. After 30 min, on each drop, growth medium was added to prevent evaporation of the drop. Then, 6 mL of PBS was added to the bottom of the plate, with the purpose of keeping the moisture level high. After 24 h, the formed spheroids were transferred to a 96 round-bottom multiwell with 200 μL of DMEM growth medium for U87 and RPMI for DU145.

### 2.6. Spheroid Dissemination Assay

The dissemination assay was performed plating spheroids on collagen matrix. Briefly, a 24 multiwell was coated with 30 μL of collagen (4 mg/mL) for 24 h before transferring the spheroids. Once transferred, the spheroids were treated with dasatinib at selected concentrations and monitored by phase contrast microscopy up to 24 h. Digital images were taken at T0 (newly transferred spheroid), T8, and T24, and were analyzed by the public domain software ImageJ (https://imagej.nih.gov/ij/ accessed on 15 September 2022) for the semiquantitative evaluation of the invaded area.

### 2.7. Cell Migration Assay

For the migration assay, cells were first seeded at a density of 5 × 10^4^ cells/mL in a 6-well multiwell and the next day were treated with dasatinib at selected concentrations for 24 h. Then, cells were detached from the culture plate by trypsin–EDTA and centrifuged at 300× *g* for 10 min. Alternatively, cells were recovered from 3D bioprinted structures with a solution of EDTA 250 mM in PBS. Cells were first counted and then seeded in serum-free medium at a density of 3 × 10^4^ cells/mL on top of the filter membrane of inserts that were placed in the 24-well multiwell. In each well was added 800 μL of complete growth medium (DMEM for U87 and RPMI for DU145), which serves as a chemoattractant. The planned incubation time was 2 h. The insert was removed from the multiwell plate, and was cleaned of the remaining cells that did not migrate from the top of the membrane. Then, 200 μL of filtered formalin was added to each insert for 10 min at room temperature, and then 200 μL of crystal violet (12 mM in a 20% solution of methanol in water) was added to each insert for 10 min at room temperature. Crystal violet was then removed, and washing with tap water was performed to remove excess dye. The cells were counted by phase contrast microscopy at 400× magnification, in at least 5 fields per membrane, to obtain the mean number of migrated cells per field.

### 2.8. Statistical Analysis

All the statistical procedures were performed by GraphPad Prism Software Inc. (San Diego, CA, USA). Data are expressed as mean ± standard deviations (SD) of at least three independent experiments. The statistical significance between measure series was calculated with parametric Student t test and *p* values of less than 0.01.

## 3. Results

*Tumor cell growth in bioprinted scaffold.* Prostate cancer (DU145) and glioblastoma (U87) cells were resuspended in alginate/gelatin hydrogel at a density of 2 × 10^6^ cells/mL. Cell suspension was bioprinted in a controlled environment onto each well of a 12-well plate following the digital model of a cuboid shape with four internal empty spaces (structure size: L 12 mm × W 12 mm × H 1.2 mm) (Figure 1a). Tumor cell growth was monitored by PrestoBlue^®^ cell viability reagent and revealed a progressive increase in the number of viable cells that was similar in the two cell lines, with a doubling time of about 14 days (Figure 1b). When bioprinted structures were observed by phase contrast microscopy, it was evident that tumor cells tended to form spheroid aggregates with increasing diameter over time (Figure 1c).

*Antiproliferative effect of dasatinib in 3D cell models*. Bioprinted tumor cells and tumor spheroids, utilizing both DU145 and U87 cells, were subjected to increasing concentrations of dasatinib (1 μM, 10 μM, and 100 μM) for 72 h and tumor growth was monitored by PrestoBlue^®^ cell viability reagent. Dasatinib determined a significant reduction in cell viability that was evident starting from 1 μM with similar trends in the two different cell models (Figure 2a,b). Dasatinib cytotoxicity was more marked in U87 cells compared to DU145 cells. In addition, according to IC50 value, bioprinted cells demonstrated a higher sensitivity to dasatinib compared to spheroids, but both spheroids and bioprints were far less susceptible to dasatinib action compared to 2D cell culture treated in the same experimental conditions (Figure 2c).

*Evaluation of Src activation*. The expression of Src and of its active form, pSrc (Tyr416), was evaluated in 2D culture and in bioprints realized with DU145 and U87 cells. Expression levels of total Src, pSrc (Tyr416), and GAPDH were evaluated in total cell lysates by Western blot analysis in 2D cell cultures treated with increasing concentrations of dasatinib (0.1, 1, and 10 μM) for 24 h. In DU145 cells, a reduction in the active form of Src, but not in the total Src, was evident already from 0.1 μM dasatinib, while in U87 cells, the first effective concentration was 1 μM (Figure 3a). Expression levels of total Src, pSrc (Tyr416), and GAPDH were also evaluated in total cell lysates by Western blot analysis in bioprint models treated with 10 and 25 μM of dasatinib for 24 h. The results did not show an appreciable reduction in the active form of Src in the DU145 model, while the first effective concentration of dasatinib in U87 cells was 25 μM (Figure 2b).

*Inhibition of cell migration by dasatinib*. Two-dimensional cell culture and bioprint models were treated for 24 h with increasing concentrations of dasatinib (0.1, 1, and 10 μM), and then cells were recovered to evaluate their migration ability. The analysis of the number of cells able to cross the membrane of the transwell system demonstrated that in DU145 and U87 cells treated in standard culture conditions, dasatinib was effective in reducing cell migration at a concentration of 0.1 μM and 1 μM, respectively (Figure 4a,b). On the contrary, cells recovered from bioprints did not show a significant reduction in cell migration with respect to control cells for DU145 cells, and only at the concentration of 10 μM for U87 cells (Figure 4b).

*Inhibition of cell dissemination in spheroids*. In order to verify the effect of dasatinib on migration of cells grown in spheroids, DU145 and U87 cells were used to form spheroids for 48 h and then spheroids were transferred to collagen-coated plates, where half of them were treated with 10 μM dasatinib. Then, spheroids were monitored by phase contrast microscopy, and time-course digital images were taken. The evaluation of dissemination was performed by measuring the area occupied by cells adhered to collagen, subtracting the area of the spheroid. The analysis of the mean relative values of the covered area revealed that cells in the control spheroids were able to disseminate progressively onto collagen from the surface of the spheroid, and this phenomenon was significantly evident starting from 8 h after cell plating (Figure 5a,b). On the other hand, dasatinib-treated spheroid demonstrated a reduced capacity to disseminate, and a significant increase in mean disseminated area was visible only after 24 h but to a smaller extent with respect to the mean control area.

## 4. Discussion

Cancer 3D tissue cultures represent an experimental model able to potentially mimic in vivo growth more closely. In particular, they show distinct characteristics in terms of cellular phenotype, mass transport, and cell–cell interactions as compared with conventional 2D cell cultures [19]. These specific characteristics can significantly affect the sensitivity to antitumoral drugs, and for this reason, it is fundamental to acquire detailed knowledge about their response in order to plan their effective utilization in preclinical drug testing. Spheroids are considered an ideal model in order to mimic some important tumor features, such as structural organization and the gradients of oxygen, pH, and nutrients [20]. Indeed, cancer spheroids have been frequently considered in preclinical studies to evaluate tumor response to chemotherapy. Although different techniques have been tested for spheroid formation, several issues with applying this model at a preclinical level still remain, particularly reproducibility and high-throughput application [21]. Recently, 3D bioprinting has been recognized as a promising technology for creating a tissue-based platform with high reproducibility and scalability. A key feature in developing a 3D bioprint model useful in pharmaceutical applications is the choice of an appropriate printable biomaterial, commonly referred to as bioink, essential for determining cancer cell phenotypes and biophysical properties of the tissue. Among the natural polymers, sodium alginate plays a significant role in tissue engineering applications owing to its biocompatibility, bioavailability, low cost, and thixotropic property [22]. Gelatin is frequently added to alginate in order to enhance cytocompatibility and printability. In our study, we confirmed that DU145 and U87 cells survive in alginate/gelatin scaffold, showing a proliferation trend for at least 14 days. Cancer cells cultured in these conditions tended to grow as cell aggregates, resembling spheroids, and this is because the bioink does not represent an adhesive substrate for cells. When we compared 3D bioprinted tissue culture with 2D culture and spheroids in their susceptibility to dasatinib treatment, we ascertained that 3D bioprinted cells behaved more similarly to the spheroid model. In fact, both 3D bioprints and spheroids demonstrated higher resistance to dasatinib toxicity than 2D cell culture. The resistance was particularly evident in DU145 cells (about 20-fold higher IC50 than 2D) and slightly higher in spheroids than in bioprinted tissue cultures (about 2-fold higher IC50 in U87).

Spheroids frequently showed in the literature high resistance to most therapies, including chemotherapies and radiotherapies [23]. The acquisition of resistance in 3D models could be explained by different mechanisms. Besides biological features, including different interactions with surrounding cells or with the extracellular matrix and higher phenotypic heterogeneity, an important element to consider is the penetration of the drug into the spheroid. Large spheroids show higher drug resistance than small ones, and the penetration of drug is restricted to the outer layer. In our experiments, we utilized spheroids of >500 μm in diameter, in which drug resistance could be attributable just to their dimensions, as mentioned before [20]. Cells in bioprinted structures formed spheroids that after 12 days are heterogeneous in diameter and <500 μm; however, as indicated by the IC50, they express a drug resistance similar to that of the spheroids. However, also in this case, the reason could be attributable to a scarce penetrance of the drug. The analysis of the activation status of Src, the main target of dasatinib, showed a reduced ability of the drug to inhibit the formation of the phosphorylated Src. Thus, it is plausible that alginate/gelatin hydrogel generates a hurdle to the diffusion of dasatinib within the bioprinted structures. Resistance to dasatinib was also evident when cells were recovered from the bioprinted structure and subjected to migration assay. In fact, with respect to the expected reduction in migration demonstrated by cells treated in 2D culture, cells from bioprinted structures maintained a higher migration ability. The drug resistance was not apparently associated with a change in phenotype. In fact, the permanence of the susceptibility to dasatinib in 3D tissue culture was confirmed by the dissemination test performed with spheroids. In this test, dasatinib was effective in reducing the ability of cells to invade the surrounding microenvironment. Because the cells involved in the dissemination are mainly those present on the surface of the spheroid, this supports the hypothesis of the scarce penetrance of the drug within the 3D structure.

The application of bioprinting to drug screening in our experimental procedures offers different advantages with respect to spheroid formation by the hanging drop method that is performed manually. In fact, the hanging drop method requires a skilled operator, and its technical complications compromise its reproducibility and make it unsuitable for high-throughput screening. However, novel 3D tissue techniques emerge constantly. These include new multi-microwell platforms able to assure higher sphere formation yield, uniform sphere size, and scalability [24]. However, in these procedures, as in all suspension cell culture methods, a reliable non-adherent coating is critical to assure reproducibility in sphere formation and in their behavior during the prolonged culture. The use of hydrogel as embedding material may guarantee that cells can organize in a stable, complex, dynamic 3D architecture making contact also with extracellular matrices. Three-dimensional bioprinting does not require particular expertise and permits obtaining several uniform structures in a short time. Alginate/gelatin hydrogel supports 3D growth of cancer cells, and as we observed also in our experimental models, cells spontaneously formed spheroids without the need for a further repulsive procedure. In addition, the bioprinted structures, for their chemical and physical stability permitting a regular change in medium without affecting cell behavior, offer the possibility to perform frequent viability tests. At the same time, alginate/gelatin hydrogel can be dissolved with ion-chelating solution, freeing spheroids that once in suspension could be utilized for further analyses. With this study, we have verified that bioprinting offers an abundant cellular material that could be easily used to analyze protein expression.

## Figures and Tables

**Figure 1 pharmaceutics-15-00372-f001:**
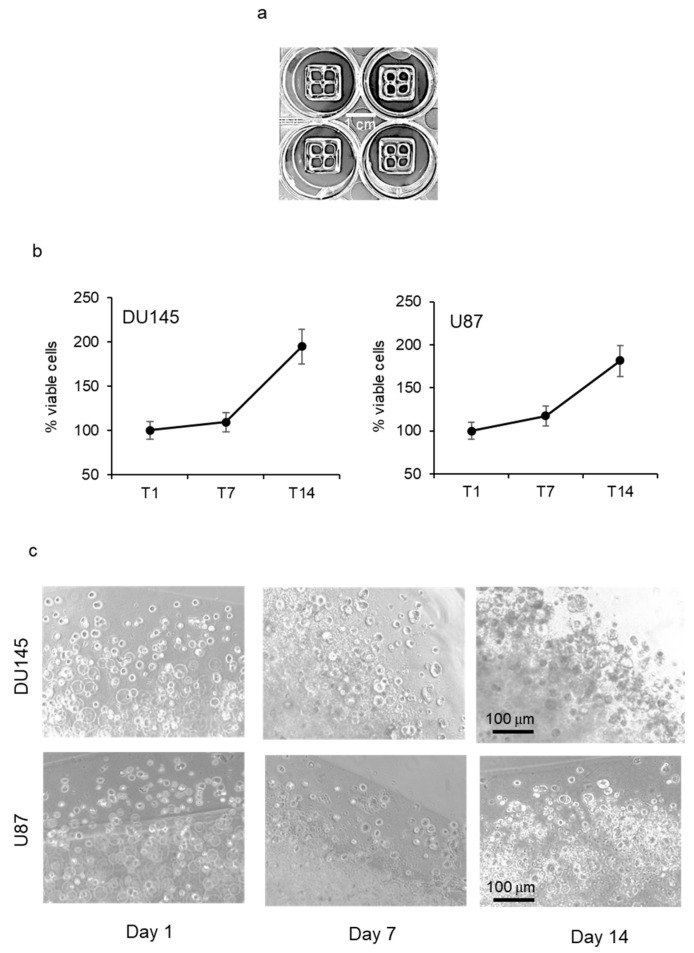
Tumor cell growth in bioprinted scaffold. (**a**) Exemplificative image of bioprinted algine/gelatin constructs printed in 12-well multiwell plate. (**b**) Mean percentage of viable DU145 and U87 cells grown in bioprinted scaffold and evaluated 1, 7, and 14 days (T1, T7, T14) after bioprinting. The mean values measured at T1 were adjusted at 100%. Each value represents the mean of five different bioprints (±standard deviation). (**c**) Representative images acquired by phase contrast microscopy (100× magnification) of bioprints containing DU145 (upper images) and U87 cells at days 1, 7, and 14 after printing.

**Figure 2 pharmaceutics-15-00372-f002:**
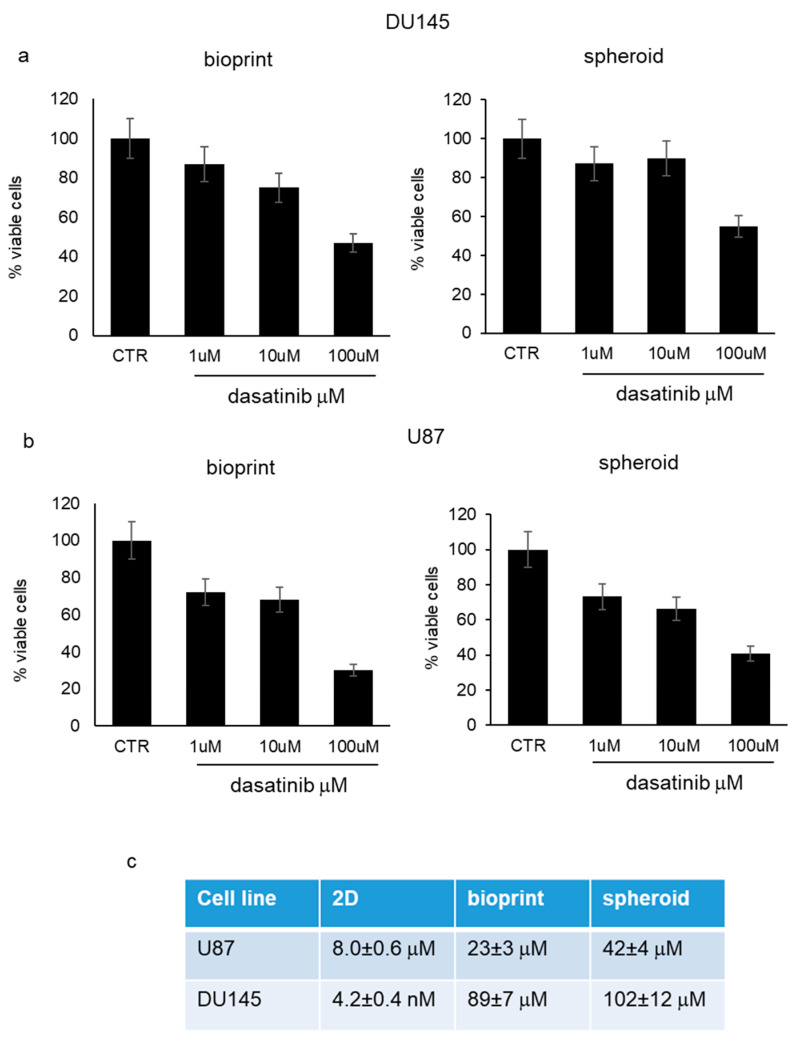
Antiproliferative effect of dasatinib in 3D cell models. (**a**) Percentage of viable DU145 cells in bioprints (**left**) and in spheroids (**right**) 72 h after treatment with increasing concentrations of dasatinib (1, 10, and 100 μM). The mean values measured in untreated cells were adjusted at 100%. Each value represents the mean of five different measurements (±standard deviation). (**b**) Percentage of viable U87 cells in bioprints (**left**) and in spheroids (**right**) 72 h after treatment with increasing concentrations of dasatinib (1, 10, and 100 μM). The mean values measured in untreated cells were adjusted at 100%. Each value represents the mean of five different measurements (±standard deviation). (**c**) Mean IC50 values calculated for U87 and DU145 cells treated with dasatinib for 72 h. The IC50s were calculated in standard culture conditions (2D), in 3D bioprint model (bioprint), and in spheroids and were the result of three different experiments (±standard deviation).

**Figure 3 pharmaceutics-15-00372-f003:**
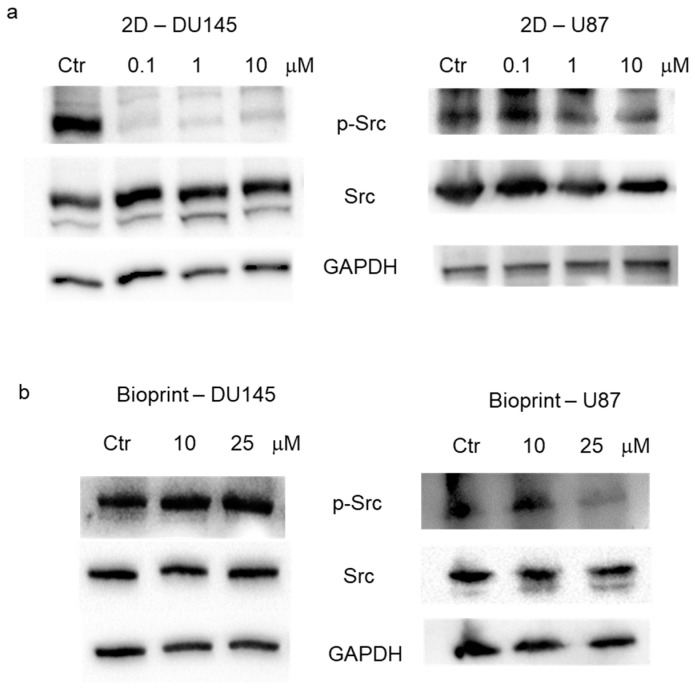
Evaluation of Src activation. (**a**) Western blot analysis of total cell lysates from DU145 (**left**) and U87 (**right**) cells cultured in standard conditions (2D). Cell lysates were recovered 24 h after dasatinib treatment with increasing concentrations (0.1, 1, and 10 μM) and analyzed for the expression of pSrc (Tyr416), total Src, and GAPDH (loading reference). (**b**) Western blot analysis of total cell lysates from DU145 (**left**) and U87 (**right**) cells cultured alginate/gelatin bioprinted scaffold (bioprint). Bioprints were treated for 24 h with 10 or 25 μM dasatinib and analyzed for the expression of pSrc (Tyr416), total Src, and GAPDH (loading reference).

**Figure 4 pharmaceutics-15-00372-f004:**
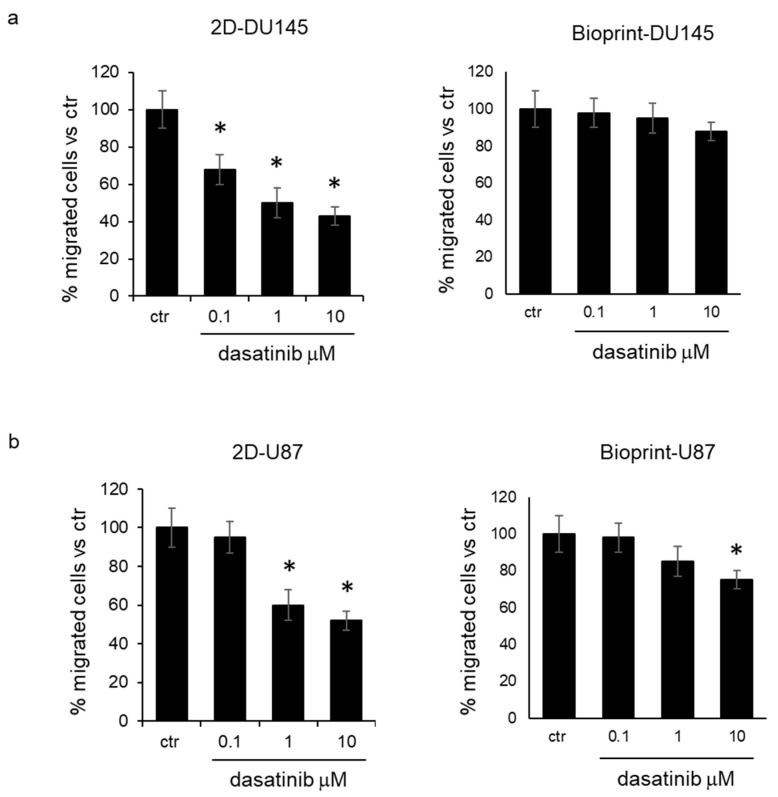
Inhibition of cell migration by dasatinib. (**a**) Percentage of migrated DU145 cells evaluated in transwell system utilizing cells from standard cell culture (2D, **left**) or from bioprint model (**right**) and treated for 24 h with increasing concentrations of dasatinib (0.1, 1, and 10 μM). The mean values measured in untreated cells were adjusted at 100%. Each value represents the mean of five different measurements (±standard deviation). * *p* < 0.01 vs. ctr, according to Student’s *t* test. (**b**) Percentage of migrated U87 cells evaluated in transwell system utilizing cells from standard cell culture (2D, **left**) or from bioprint model (**right**) and treated for 24 h with increasing concentrations of dasatinib (0.1, 1, and 10 μM). The mean values measured in untreated cells were adjusted at 100%. Each value represents the mean of five different measurements (±standard deviation). * *p* < 0.01 vs. ctr, according to Student’s t test.

**Figure 5 pharmaceutics-15-00372-f005:**
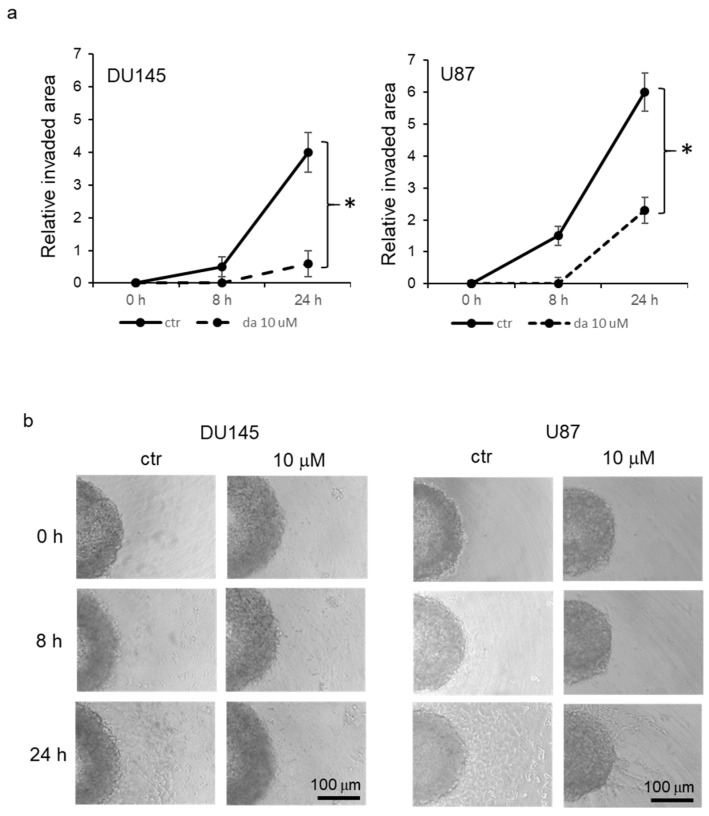
Inhibition of cell dissemination in spheroid. (**a**) Mean relative invaded area in DU145 and U87 spheroids evaluated 8 and 24 days after plating onto collagen-coated surface. The values represent the mean of five different measurements (±standard deviation) of spheroids treated or not with 10 μM dasatinib. * *p* < 0.01 between indicated points according to Student’s *t* test. (**b**) Representative images acquired by phase contrast microscopy (100× magnification) of DU145 and U87 cells at 0, 8, and 24 h after plating in the presence or not of 10 μM dasatinib.

## Data Availability

Most data generated or analyzed during this study are included in this article. The datasets and materials used and/or analyzed during the current study are available from the corresponding author on reasonable request.

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
