# Peer review of "Comparative Analysis of Dasatinib Effect between 2D and 3D Tumor Cell Cultures"

_pharmaceutics, 2023, doi:10.3390/pharmaceutics15020372_

Round 1

Reviewer 1 Report

Manuscript entitled"Comparative analysis of dasatinib effect between 2D and 3D tumor cell cultures" by Samantha Sabetta et al is about comparison between 2D and 3D cell cultures use in tumor research. Authors have used Dasatinib as reference material and validated their assays in 2D and 3D cell cultures. Authors have introduced 3D printing using sodium alginate and gelatin combination to test their hypothesis.

Minor comments:

Introduction: Introduction was more lengthy and  describes more about the Dasatinib compound. Authors should have provided more introduction for 2D and 3D cell cultures used for tumor research and it's advantages and disadvantages followed by brief introduction about Dasatinib.

Otherwise, research design and detailed methodologies were described well. Authors have discussed their results in detail.

Minor spell check may be required, like Page#7, Line#269 "citotoxicity" instead of cytotoxicity.

Overall this manuscript is good and well presented.

Author Response

We thank the reviewer for their detailed evaluation, the helpful suggestions.  A detailed point-by-point response to the comments is provided below:

Minor comments

  1. Introduction: Introduction was more lengthy and describes more about the Dasatinib compound. Authors should have provided more introduction for 2D and 3D cell cultures used for tumor research and its advantages and disadvantages followed by brief introduction about Dasatinib.

Response: The introduction was largely rewritten eliminating redundant parts and adding a new section on dasatinib (line 71-85). In addition, we have provided more details in the section that describes differences among 3D models (line 99-111; 129-140). References have been changed accordingly.

  1. Minor spell check may be required, like Page#7, Line#269 "citotoxicity" instead of cytotoxicity.

Response: we have checked the text for spelling and language rules

Reviewer 2 Report

The paper “Comparative analysis of dasatinib effect between 2D and 3D tumor cell cultures” by Sabetta et al compares the drug efficacy of 2D and 3D cell cultures. The comparison is interesting but overall lacks novelty. It is recommended the article should be significantly revised before further consideration.

1.    There are many prior publications showing the 3D culture of tumor cells and comparison with 2D cell culture. It is recommended that the authors should better discuss the pros and cons of existing platforms and comparison.

https://www.ncbi.nlm.nih.gov/pmc/articles/PMC5754907/

https://www.ncbi.nlm.nih.gov/pmc/articles/PMC4495468/

https://www.nature.com/articles/srep21061

https://www.mdpi.com/1422-0067/19/1/181

https://www.sciencedirect.com/science/article/pii/S092540051830248X?casa_token=_PGmtXikZ4QAAAAA:4lLNKVsA2RnN5SPtNhLpuH6pCdAib18SffS1ZPdU6g19VTNPSWt7LII5KrDWN_-bXWfl85Q3xw

2.    In addition to publications, there are many commercial products providing the capability of 3D culture. Some examples are attached below. The authors should better articulate the presented method in the last paragraph of Discussion.

https://ibidi.com/content/379-controlled-cell-adhesion-with-ibidi-patterning,

https://facellitate.com/product/biofloat-384-well-plates/

https://www.sigmaaldrich.com/US/en/technical-documents/technical-article/cell-culture-and-cell-culture-analysis/3d-cell-culture/high-throughput-truegel-3d-hydrogel-plates#HydrogelPlates

3.    It is recommended that the authors can confirm the exchange of nutrients through the hydrogel and bioprinting toward cells.

4.    While the authors mentioned “high-throughput” multiple times, only a few experiments have been tested. It is not convincing the presented method is high-throughput as compared to many prior publications and commercial products.

5.    The image quality of Fig. 1 is not great. It is hard to see spheroids.

Author Response

We thank the reviewer for their detailed evaluation, the helpful suggestions.  A detailed point-by-point response to the comments is provided below:

The paper “Comparative analysis of dasatinib effect between 2D and 3D tumor cell cultures” by Sabetta et al compares the drug efficacy of 2D and 3D cell cultures. The comparison is interesting but overall lacks novelty. It is recommended the article should be significantly revised before further consideration.

Response: other studies have previously demonstrated that spheroids are more resistant to different drugs than 2D culture. In our study we aimed at verify this information for a relatively new technique, such as alginate bioprinting, and at comparing cell response to dasatinib between bioprinted model and spheroid model. We think that our results could be useful for scientists mainly interested in evaluating the use of alginate bioprinting in cancer models. 

1.+2    There are many prior publications showing the 3D culture of tumor cells and comparison with 2D cell culture. It is recommended that the authors should better discuss the pros and cons of existing platforms and comparison. In addition to publications, there are many commercial products providing the capability of 3D culture. Some examples are attached below. The authors should better articulate the presented method in the last paragraph of Discussion.

Response: we agree with the reviewer. The abundance and heterogeneity of techniques available in literature render impossible to be exhaustive in this context. However, we have expanded the description of the different culture models in the introduction, mainly those pertinent our study, and have discussed with further details the pros and cons respect to the models we used in our experiments (INTRO: line 99- ; Discussion: line 410- ) taking into account some new refes suggested by reviewer.

  1. It is recommended that the authors can confirm the exchange of nutrients through the hydrogel and bioprinting toward cells.

Response: several studies have utilized alginate/gelatin hydrogel because its biocompatibility and chemical/physical properties and show elevated capacity of diffusion of the medium through this hydrogel. In addition, alginate gels have been investigated for the delivery of a variety of drugs, demonstrating the rapid diffusion of small molecules through the gel ( see ref: Lee KY, Mooney DJ. Alginate: properties and biomedical applications. Prog Polym Sci. 2012 Jan;37(1):106-126. doi: 10.1016/j.progpolymsci.2011.06.003. PMID: 22125349; PMCID: PMC3223967.)

  1. The image quality of Fig. 1 is not great. It is hard to see spheroids.

RESPONSE: figure contrast has been enhanced and high-res versions of the photos have been submitted

Round 2

Reviewer 2 Report

The authors addressed the comments from the reviewer.